# Mycobacterium Tuberculosis and Avium Complex Investigation among Malaysian Free-Ranging Wild Boar and Wild Macaques at Wildlife-Livestock-Human Interface

**DOI:** 10.3390/ani11113252

**Published:** 2021-11-13

**Authors:** Yusuf Madaki Lekko, Azlan Che-Amat, Peck Toung Ooi, Sharina Omar, Siti Zubaidah Ramanoon, Mazlina Mazlan, Faez Firdaus Abdullah Jesse, Sabri Jasni, Mohd Firdaus Ariff Abdul-Razak

**Affiliations:** 1Department of Veterinary Clinical Studies, Faculty of Veterinary Medicine, Universiti Putra Malaysia, Serdang 43400, Selangor, Malaysia; ymlekko@unimaid.edu.ng (Y.M.L.); ooi@upm.edu.my (P.T.O.); jesse@upm.edu.my (F.F.A.J.); 2Department of Veterinary Medicine, Faculty of Veterinary Medicine, Universiti of Maidugu-ri, Maiduguri PMB 1069, Borno State, Nigeria; 3Department of Veterinary Pathology & Microbiology, Faculty of Veterinary Medicine, Universiti Putra Malaysia, Serdang 43400, Selangor, Malaysia; sharina@upm.edu.my (S.O.); m_mazlina@upm.edu.my (M.M.); 4Farm and Exotic Animal Medicine and Surgery, Faculty of Veterinary Medicine, Universiti Putra Malaysia, Serdang 43400, Selangor, Malaysia; sramanoon@upm.edu.my; 5Department of Paraclinical, Faculty of Veterinary Medicine, Universiti Malaysia Kelantan, Kota Bharu 16100, Kelantan, Malaysia; jasni@umk.edu.my; 6Department of Wildlife and National Parks Peninsular Malaysia, Kuala Lumpur 56100, Selangor, Malaysia; mfirdaus@wildlife.gov.my

**Keywords:** *Mycobacterium tuberculosis* complex, *Mycobacterium avium* complex, polymerase chain reaction, post mortem lesion, Selangor, tuberculosis, wild-life-livestock-human interface

## Abstract

**Simple Summary:**

This study targeted a small epidemiological area of a selected wildlife-livestock-human interface in Selangor to detect important veterinary and public health mycobacteria in free-ranging wild boar (*Sus scrofa*) and wild macaques (*Macaca fascicularis*) using a combination of diagnostic methods, tuberculosis-like lesion (TBLL) detection and nucleic acids detection by conventional and molecular analyses. Conventional PCR on wild boar tissues showed that 75% (9/12) of the lymph node samples were positive for *Mycobacterium bovis* (95% CI: 46.8–91.1). For macaques, 33.3% (10/30) were positive for *Mycobacterium avium* (95% CI: 19.2–51.2).

**Abstract:**

Wild animals are considered reservoirs, contributing to the transmission of emerging zoonotic diseases such as tuberculosis (TB). A cross-sectional study was conducted by opportunistic sampling from fresh carcasses of free-ranging wild boar (*n* = 30), and free-ranging wild macaques (*n* = 42). Stained smears from these tissues were tested for acid-fast bacilli (AFB) with Ziehl–Neelsen staining. Mycobacterial culture was conducted using Lowenstein–Jensen media and Middlebrook 7H11 agar media. Polymerase chain reaction (PCR) was performed through the detection of the 16S rRNA gene, with multiple sets of primers for the detection of *Mycobacterium tuberculosis* complex (MTBC) and *Mycobacterium avium* complex (MAC). In wild boars, 30% (9/30; 95% Confidence Interval: 16.7–47.9%) of examined samples showed gross tuberculosis-like lesions (TBLLs). Multiple nodular lesions that were necrotic/miliary with cavitation were found in the submandibular lymph nodes, tonsils, lungs, kidney and liver, while single nodular lesions were found in the mediastinal lymph nodes, spleen and mesenteric lymph nodes. Conventional PCR on the submandibular lymphoid tissues of wild boar (nine samples with TBLLs and three non-TBLL samples) showed that 75% (9/12) were positive for *Mycobacterium bovis* (95% CI: 46.8–91.1), and 91% (CI: 64.6–98.5) were positive for *Mycobacterium avium*. For macaques, 33.3% (10/30) were positive for *M. avium* (95% CI: 19.2–51.2) but negative for MTBC.

## 1. Introduction

*Mycobacterium tuberculosis* complex (MTBC) members are responsible for tuberculosis (TB) in domestic and wild animals and humans. Wild boar may facilitate the epidemiology of TB infection by acting as true maintenance hosts, which means the infection can persist without external sources or spillover hosts, such as bovine TB in livestock [1]. Wild boar that share grazing and water resources with other livestock can complicate the epidemiology of bovine TB [2]. *Mycobacterium avium* complex (MAC) infection is ubiquitous in nature and is distributed in the environment, especially in soil, marshland, rivers and streams [3]. Infection of MAC in free-ranging wild boar with or without clinical signs had been reported [4,5]. Therefore, it is imperative to survey both infections of MTBC and MAC that might be circulating in our free-ranging wild boar using opportunistic sampling of carcasses at the wildlife-livestock-human interface (WLHI). The WLHI represents a critical point for cross-species transmission and emergence of pathogens [6]. The state of Selangor (sampling area) is a typical example, which has a number of high-risk areas for zoonotic TB in domestic cattle and human, due to the high number of dairy cattle farms and the diversity of wildlife and encroachment of wildlife into human settlements.

The inadequacy in research findings on the full NHP pathogen spectrum results in the lack of understanding of *Mycobacterium* that is associated with TB in non-human primates (NHP) [7]. It was reported that TB in both captive and free-ranging NHP was caused by *M. tuberculosis*, and occasionally by *M. bovis* and *M. africanum*, and thus has led to a presumption that free-ranging NHP could also be susceptible to TB. Indeed, NHP may become infected with MTBC and develop the disease, but more studies are needed to understand the epidemiology of the disease in this species [8]. Non-human primates are also susceptible to an important non-tuberculous mycobacteria, namely, MAC infection associated with Johne’s disease or paratuberculosis caused by *M. avium* subsp. *paratuberculosis* (MAP) [9]. Members of MAC are not species-specific and are frequently associated with animal or human diseases that resemble tuberculous-like lesions in lymph nodes and parenchymatous organs [10]. Various outbreaks of MAC infection have been reported in humans and NHP worldwide [9,10,11]; however, the relevance of free-ranging NHP for MAC transmission among animals and to humans is unclear [12,13,14].

Many wild animals have moderately high ecological interactions with people, which means they share the same environments with people, potentially causing conflicts. The WLHI ensues in the form of damage to planted crops, livestock, livelihoods and disruptions to the wildlife itself, with cost implications for both sides. The moderation of the WLHI is complicated and requires more than just reducing the amount of damage produced by wildlife but also the reduced risk of disease transmission [15,16,17]. There is a paucity of research studies on wildlife TB and other mycobacterial infections in free-ranging wild boar and macaques in Malaysia, despite the abundance of these animal species [18], and their potential as the maintenance hosts for these infections cannot be neglected.

The diagnosis of TB in wildlife species [19,20] often depends on post-mortem, acid fast stain, microbial culture and PCR tests. Mycobacterial culture is the gold standard method, but it mainly depends on the growth rate of MTBC bacteria, which normally takes several weeks to grow and therefore delays prompt diagnosis and treatment [21]. To hasten and improve diagnosis, nucleic acid amplification tests (NAAT) were produced and developed for rapid TB diagnosis. The NAAT techniques have improved TB diagnosis compared to mycobacterial culture [22]. PCR makes use of nucleic acid amplification to exponentially replicate specific target genes. Its sensitivity and specificity make PCR one of the most used techniques in the diagnosis and study of MTBC [23].

Although there is significant improvement in molecular diagnostic techniques towards the development of faster and more accurate detection for MTBC in human samples, only few techniques had been described for detecting MTBC directly from animal tissue samples, especially fresh tissues from livestock and wild animals [23]. From a local perspective, wildlife could be a potential host, circulating and transmitting important mycobacterial infections among animals and humans. This surveillance study targeted a small epidemiological area of a selected wildlife-livestock-human interface in Selangor to detect the important veterinary and public health mycobacteria in free-ranging wild boar (*Sus scrofa*) and wild macaques (*Macaca fascicularis*) by a combination of diagnostic methods such as tuberculosis-like lesion (TBLL) detection, acid fast staining, mycobacterial culture and antigen detection by conventional and molecular analyses.

## 2. Materials and Methods

### 2.1. Study Area and Animals

This research complied with the Department of Wildlife and National Parks (PERHILITAN) regulations with research permits issued (B-00156-15-19 for free-ranged wild boar, *Sus scrofa*; W-00396-15-19 for free-ranged wild macaques, *Macaca fascicularis*) and institutional approval of animal ethics (UPM/AICUC/AUP-U040/2019). Cross-sectional sampling was conducted from April to August 2019–2020 by opportunistic sampling of fresh carcasses of free-ranged wild boar (*n* = 30), and samples were collected by registered pleasure hunters and wildlife department officials at the WLHI within the state of Selangor (central Peninsular Malaysia). These legal hunters were licensed by the PERHILITAN, and hunting was irregular, not fixed, and was self-managed, as it was part of their social activity. The ages of the wild boar were determined [15] and categorized as 4 yearlings, 9 sub-adults and 17 adults. They were harvested from the following localities: Sepang (*n* = 13), Kuala Selangor (*n* = 8), Hulu Selangor (*n* = 4), Kuala Langat (*n* = 3) and Kajang (*n* = 2). Carcasses of free-ranged wild macaques (*n* = 42) (5 yearlings, 11 subadults and 26 adults) were collected from different areas within Selangor, where these macaques were harvested from the population control activity by the PERHILITAN at the WLHI areas.

### 2.2. Sample Collection and Post-Mortem Examination for TBLL

Blood samples collected from carcasses of wild boar and wild macaques by venipuncture of the cavernous sinus [24] were transferred into plain and EDTA tubes. The serum was extracted and kept at −20 °C for further analysis. Post-mortem was conducted on the fresh carcasses of wild boar and wild macaques. The organs collected were tonsils, submandibular lymph nodes (LN), lungs, tracheobronchial and mediastinal LN, spleen, liver, kidney and mesenteric LN. These organs were examined macroscopically for the presence of TBLL, categorized based on the referral descriptions [25] as type 1—necrotic/miliary tuberculosis; type 2—pleuritis/peritonitis; type 3—caseous; type 4—cavitation; type 5—calcification/mineralization; and type 6—purulent lesion. Tissues collected were stored at −20 °C for further analysis.

### 2.3. Stained Smears for Acid-Fast Bacilli (AFB)

Contact smears of every tissue collected were made on microscopic slides and stained with the Ziehl–Neelsen method. Each slide was examined across the entire length of the smear at 1000× amplification using immersion oil for about 10 min to record the presence or absence of AFB [26].

### 2.4. Mycobacterial Culture Isolation

Decontaminating agents were prepared by dissolving 4.0 g of sodium hydroxide (NaOH) pellets in 100 mL of sterile distilled water to make a final concentration of 2% NaOH [27]. For culture, briefly, 1 to 5 g of tissues macerated in 80 mL of sterile distilled water using a stomacher. To the homogenized tissue, 2%NaOH was added, and the suspension was vortexed for proper mixing and then kept at the 37 °C for 30 min; thereafter, it was neutralized with 10% sulfuric acid (H_2_SO_4_) using litmus paper as indicated at pH 7. The suspended tissues were centrifuged at 5600 rpm for 20 min, and the pellet sediments were used for inoculation on Lowenstein–Jensen (L–J) media and Middlebrook 7H11 using sterile swab sticks, and incubated at 37 °C in CO_2_. They were examined every week, up to a period of 15 weeks, to detect the growth of colonies with the characteristic dry, rough texture and the cream color of MTBC.

### 2.5. DNA Extraction from Tissues and Anticoagulated Blood

Extraction of DNA from tissues and blood using the DNeasy^®^ Blood and Tissue Kit (Qiagen^®^, Hilden, Germany) was based on the protocol by the manufacturer.

### 2.6. PCR for the Genus Mycobacterium, MTBC and MAC

Conventional PCR on 12 lymphoid tissue samples of wild boar, and 10 tissue samples of macaques was conducted. The oligonucleotide primers used in this study were manufactured by Integrated DNA Technologies, Inc., USA, distributed by Apical Scientific Sdn Bhd Malaysia (refer to Table 1). The primer MYCGEN-F and MYCGEN-R generate 1030 bp for genus detection common to MTBC members. MYCAV-R and MYCGEN-F generate 180 bp for detection of *M. avium*. MYCINT-F and MYCGEN-R generate 850 bp for detection of *M. intracellulare.* TB1-F and TB1-R generate 372 bp for *M. bovis* detection, oxyRF, oxyRR, L1 and L2 for *M. bovis* detection in wild boar [28]. Primers Rv2073cF and Rv2073cR for *M. tuberculosis* (*M. tb*) were used for detection in tissue samples of 10 macaques, while hsp65F and hsp65R were for detection of the genus common to MTBC members [28,29]. The primers were aliquot based on specified protocols by the producers. The primers were centrifuged for a short time to spin down all the powder. TE buffer was added according to each desired volume that was stated on the tube. The solution was mixed by pipetting using a 100 µL micropipette. The concentration of the solution became 100 µM. Then, 10 µL of the concentration was mixed with 90 µL TE buffer to make 10 µM. Polymerase chain reaction was carried out based on the referral protocol [28]. Briefly, 25 µL reaction mixture was prepared containing 12.5 µL TopTaq Master Mix 2x (Qiagen^®^, Germany), 5.5 µL RNase free water, 5µl DNA template and 1 µM of each primer set. The positive control used was *Mycobacterium avium* subspecies *avium* Chester (ATCC^®^ 15769^TM^), and the negative control was RNase free water (Qiagen^®^, Germany). The amplification conditions were as follows: initial denaturation at 94 °C for 5 min of 39 cycles each and final denaturation of 30 s, annealing at 62 °C for 3 min, extension at 75 °C for 3 min and final extension at 75 °C held for 5 min [28].

For the wild macaques, PCR was carried out as per previous protocols [28,29]. Briefly, a 25 µL reaction mixture was prepared containing 12.5 µL TopTaq Master Mix 2x, 5.5 µL RNase free water, 5 µL DNA template and 1 µM of each primer (refer to Table 1). Primers amplified the 556 bp, 460 bp, 600 bp and 441 bp fragments targeting *oxyR*^285^, 16S rRNA, Rv2073c (RD9) and *hsp65*^631^ gene, respectively. Positive and negative controls used were *M. tuberculosis* H37R3 (ATCC^®^ 25177^TM^) and RNase free water, respectively. The amplification conditions were initial denaturation at 94 °C temperature for 5 min followed by 25 cycles final denaturation at 94 °C for 1 min, annealing at 60 °C for 1 min and initial extension at 72 °C for 1 min, and final extension at 72 °C for 10 min. As for L1 and L2 amplification, initial denaturation at 95 °C for 5 min, followed by 30 cycles final denaturation at 95 °C for 1 min, annealing at 65 °C for 1 min, initial extension at 72 °C for 1 min and final extension at 72 °C held for 10 min. Gel electrophoresis was performed using TBE buffer and stained with FluoroSafe DNA stain, and 5 µL of PCR product was mixed with 2 µL loading dye prior to loading it into the 5 mm wells. The PCR products were subjected to 80 V for 2 h on a 2% agarose gel and were viewed using Alphamager^TM^ (Alpha Innotec, Germany).

### 2.7. DNA Sequencing and Phylogenetic Analysis of M. bovis 16S rRNA Gene

The DNA sequencing were conducted by First BASE Laboratories, Malaysia. Sanger methods were used to sequence each purified DNA. The sequencing results of the 16S rRNA gene were edited using the BioEdit software and aligned using molecular evolutionary genetics analysis software MEGA X. The basic local alignment search tool (BLAST) was used to locate regions of similarity between biological sequences. The edited partial 16S rRNA gene sequences were compared to those available on the BLAST website to confirm that these sequences were similar to the *M. bovis* 16S rRNA gene in which sequence similarity was identified and noted. The 16S rRNA gene sequence of 8 reference strains from other countries that had similarity with our sequences were downloaded and aligned with sequences from our samples. Immediately upon sequence alignment, parts of the sequences were cut to ensure that the length of base pairs of any sequences were similar to enable for precise phylogenetic tree generation. The phylogenetic tree was constructed by using the UPGMA tree method using MEGA X software. Tree reliability was tested using 1000 bootstrap replications, and the substitution type was a nucleotide. The phylogenetic tree construction was carried out on the bases of 16S rRNA gene sequences from Selangor and the reference sequences from other countries.

### 2.8. Statistical Analyses

Epitools epidemiological calculators were used to calculate apparent prevalence at a confidence interval of 95% [30].

## 3. Results

In wild boar, 30% (9/30; 95% Confidence Interval: 16.7–47.9%) of examined samples showed gross TBLL (Table 2). Among the animals with lesions, one was a piglet (11.1%), two were sub-adults (22.2%), five were adults (55.5%) and the age of the other one was unknown (incomplete carcass). Eight wild boar had type-1 necrotic/miliary TBLL, and only one had type-4 TBLL cavitation (Figure 1). The TBLLs were not specific to one organ tissue. They were observed in the submandibular LN, tonsil, lung, mediastinal LN, liver, spleen and kidney. No TBLL were found in any macaques. Contact smears of tissues of wild boar and macaques stained with Ziehl–Neelsen staining showed no detection of AFB. Culture of tissues from wild boar and macaques on L–J media and Middlebrook 7H11 agar were considered negative for MTBC after 15 weeks of incubation.

PCR results are presented in Table 2 and Figure 2. For wild boar, PCR was only done for 12 wild boar (9 TBLL and 3 non-TBLL), and 75% (9/12) at (95% CI: 46.8–91.1) of these samples were positive for *M. bovis*, while 91% (64.6–98.5) of the sample were positive for *Mycobacterium avium*. PCR from macaques were also negative for MTBC nucleic acids, but *Mycobacterium avium* were detected at 33% (10/30) (95% CI: 19.2–51.2). *M. bovis* nucleic acids was detected with primer TB1-F and TB1-R at 372 bp and *M. avium* nucleic acids was detected with primer MYCAV-R and MYCGEN-F at 180 bp (see Table 2 and Figure 2).

The amplified fragments of the DNA product sequence result showed that the Malaysian strain 3653770 W1A TB UPM had an identified similarity index of 100% with *M. tuberculosis* variant *bovis* strain 1 and query coverage of 99% (E value 7e–168 with 100% identical, accession CP040832.1) (Figure 3).

## 4. Discussion

This is the first surveillance of MTBC and MAC among Malaysian free-ranging wild boar and wild macaques in the wildlife-livestock-human interface. The detection rate of wild boar that showed gross pathology was 30%, while higher values were reported at 82.6% by [31], 75% [32] and 72.2% [26] in wild boar. These high prevalence areas have high densities of wild boar and intensive game-management, including fencing, supplementary feeding and translocations. In contrast, wild boar hunting game is uncommon in Malaysia, and this may contribute to the low sample size and prevalence observed in the study. Gross TBLL were found in three anatomic body areas, namely the head (predominantly in submandibular LN and tonsil), the thorax (lungs) and abdomen (kidney and liver), which can either be localized or generalized, as reported in other studies [31,33,34]. Localized lesions could be due to infection transmitted through the oral route by scavenging of tuberculous carrion such as TB-infected dead animals [35], while generalized lesions could be attributed to either the respiratory or digestive infection route [31]. In this regard, the local wild boar may obtain the infection through contact with carrion from the same species, other potential wildlife, direct or indirect spillover from livestock with TB, or MTBC-contaminated environments, such as in soil and water.

Lesions found in submandibular LN, lung, mediastinal LN, tonsil, spleen, kidney and liver appeared necrotic and miliary in nature, while one appeared in the form of cavitation, which is an enlarged liquefying lesion, obtained from the submandibular LN, which resembled those reported previously [6,31,33]. TB-like lesions were found more in adult and sub-adult wild boar with a high percentage of 78%, which is in agreement with other findings [36]. This could be explained by the fact that the adults and sub-adults have a well-developed immune system that allows them to withstand chronic TB disease compared to piglets, thus the lesions are more severe in advanced disease [37,38]. Lesions were found more in submandibular LN in our samples and confirmed the reports that the TBLL from the post-mortem examination of the submandibular LN are adequate for the detection of more than 90% of TB cases (confirmation by culture positive), and this is related to the direct ingestion of mycobacterium organisms from feed [31].

Contact smears made from all the tissues and stained with Ziehl–Neelsen in this study yielded negative results. Smear sensitivity varies greatly based on the AFB burden, and because of its low sensitivity and specificity, it requires about 5000 to 10,000 cfu/mL for reliable detection [21,39]. A similar situation occurred where mycobacteria were not detected in the organs that were submitted for microscopic evaluation after the Ziehl–Neelsen test on the submandibular LN [34], while in another study, a large number of granulomas did not produce AFB following Ziehl–Neelsen staining in wild boar [40,41]. This is because the cell wall of mycobacterium is composed of a high concentration of complex lipids, which were elongated chains of fatty acids referred to as mycolic acids [42,43]. These mycolic acids cause the cell wall to be hydrophobic with increased resistance to desiccation, killing by disinfectants and staining with basic aniline dyes, and they prevented the efficacy of drug used for treatment of mycobacterial disease [21]. Absence of AFB from typical tuberculous lesions was not evident enough to rule out a post-mortem diagnosis of MTBC [44]. During the planning of our research, considering our budget constraint, we decided to conduct either stained smear (AFS) or histopathology. We later decided to use stained smear (AFS) using Ziehl–Neelsen staining, which is the method used in low resource settings, instead of histopathology to diagnose tuberculosis.

All samples were cultured on Lowenstein–Jensen and Middlebrook 7H11 agar, which were considered negative for MTBC after 15 weeks of incubation. Although culture is considered the standard bacteriological method for MTBC diagnosis [27], the sensitivity of culture however is affected by factors that affect primary isolation of MTBC in tissue samples. These factors include sample storage, period of incubation, methods of decontamination used, type of media and the fact that fibrotic and calcified granulomas may not contain viable bacteria [45,46,47]. The storage of samples is vital for the cell and reduces the rate of contamination by microbes. Normally, sample processing should commence immediately after collection, and the storage and transportation of samples after collection at 4 to 6 °C increases recovery rates when sunlight is avoided [48].

Samples maintained in a cold chain should be processed for culture within a short period of 24–48 h after collection [47]. Decontaminants have adverse consequences, such as increased time for the appearance of colonies and reduced number of recovered colonies. High loss of cell viability and increased time for colony appearance were observed when NaOH was used, and the toxic effects of these chemicals resulted in death and damage to the survivability of the bacilli [27]. There have been reports of the early appearance of colonies on agar-based media than in egg-based media, while the increased amount of colony and more positive growth were observed on egg-based media. Such differences were attributed to different contamination rates [27]. Aspects of bacteriological diagnosis, such as temperature, duration of storage and decontamination, remain to be fully appraised in order to draw conclusions on their effects on the sensitivity of bacterial culture [27].

When PCR was only performed on affected organs or lymphoid organs (visible lesions 9 and 3 non-TBLL), 75% (9/12) were positive for MTBC. PCR on a smaller sample volume carries a few risks that might lead to false negative results. These setbacks may be due to non-homogeneous distribution of mycobacterium bacilli in the affected tissues, the presence of non-visible lesions and the assumption that lesions from chronic infections have undergone sterilization [49,50]. This made it difficult to detect the presence of mycobacteria and their location in the organ to permit the targeted DNA extraction of MTBC [50]. When PCR was performed on wild boar tissues containing few mycobacteria, detection was poor due to the difficulty of amplifying mycobacterial DNA from tissues having a large amount of eukaryotic DNA [26]. Generally, the steps employed before PCR, such as sample processing and extraction of DNA can have impact on the capability to detect and measure genomes of bacteria in tissue samples [51].

The sequence sample 3653770 W1A TB UPM showed a 100% similarity index with CP040832.1: *M. tb* variant *bovis* strain1 chromosome from Brazil. High genetic diversity among *M. bovis* strains circulating in Brazil, and animals, especially livestock, could be exported from this country to other parts of the world [52]. The important factor to consider for the presence of the infection are the animal husbandry systems, animal movement between regions, the presence of wildlife reservoirs and sharing of pastures and water sources [2,53]. Sequence analysis can be a useful epidemiological tool to understand the diversity, spread, geographical localization, host preference and worldwide distribution of mycobacteria [54].

*Mycobacterium avium* was detected by PCR from wild boar, including in other studies [3,55]. *Mycobacterium avium* infection can occur with or without lesions in wild boar [4]. The porcine and human isolates of *M. avium* subsp. *hominissuis* isolates showed similarity of over 90% [56]. This finding indicated that local wild boar are important for possible *M. avium* transmission. MAC members had been extracted from blood, lungs and spleen by PCR [57]. The extraction of MAC members from the blood showed how ubiquitous these environmental bacteria could be. Many studies showed that *M. avium* positive samples by PCR were detected from tissues of macaques [9,10,58,59]. For this reason, epidemiological studies about MAC are important for public health concerns [60].

*Mycobacterium avium* complex are known to cause a variety of diseases including ‘tuberculosis-like’ diseases in humans, domestic and wild animals [61]. This is because they are known to survive on a wide variety of sources such as soil, water and foodstuffs, and through contamination from domestic and wild animals. Environmental mycobacteria of the members of MAC constitute a very interesting group in terms of ecology. They are isolated in all types of water sources and are able to produce biofilms because they can survive for a long time in the environment. In addition, they are found in fresh or frozen fruits and vegetables, contaminated feedstuffs, bedding materials, soil and saw dust, and these are identified as natural reservoirs of MAC [62]. The existence of *M. avium* in wildlife species and the likelihood of interspecies transmission may have significant effects for the control of TB and may hamper bovine TB eradication programs [59]. Management approaches such as fencing will help separate wildlife and livestock from common resources such as waterholes or feeding sites, and also the use of training dogs will help reduce wildlife visits to farms. Wildlife animals shot by hunters in endemic areas must be removed (including viscera and their remains) in order to reduce infection spread. Non-disposal of carcasses and hunting remains has contributed to wildlife disease–related conflict between hunters, government agencies and conservationists. The use of personal protective clothing, sanitizers and disinfectants after slaughter by hunters will help in reducing the spread of TB to domestic livestock.

## 5. Conclusions

Gross pathological TB-like lesions and PCR-positive results suggest that natural TB infection exists in free-ranging wild boar populations in selected areas in Selangor, and wild boar could be a spillover and/or spillback host of TB transmission to livestock and other wildlife animals. However, their role as a true reservoir needs further investigation. The significant detection of MAC among wild and captive animals is important to review the control measures, as it might cause serious issues when the animal is under low immunity. TB tests in most wildlife are still not validated or have low sensitivity; thus, the diagnosis of TB should be interpreted with care, especially when there is cross-detection with other mycobacteria, such as MAC, which may compromise the result. Thus, further investigation is warranted to link the role of multispecies interaction among free-ranging wildlife and livestock in the MTBC and MAC transmission.

## Figures and Tables

**Figure 1 animals-11-03252-f001:**
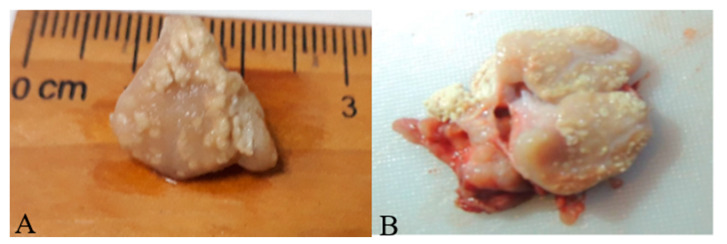
Type-1 TBLL necrotic/miliary (**A**) and type-4 TBLL cavitation lesion in submandibular lymph node (**B**) in adult wild boar.

**Figure 2 animals-11-03252-f002:**
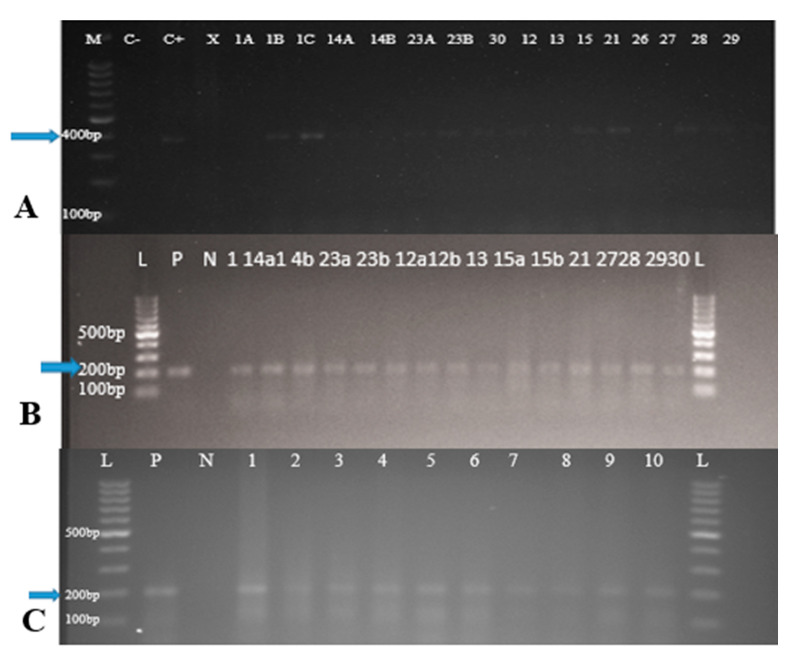
(**A**) Results of MTBC-specific PCR from organ tissues of wild boar with TBLL only. Lane M: 100 bp size ladder; C–: Negative control, C+: Positive control, X: clinical sample not related to this study. Positive samples 1 (**A**,**B**), 12, 14 (**A**,**B**), 15, 21, 23 (**A**,**B**), 27, 28, 30. (**B**) Results of MAC-specific PCR from organ tissues of wild boar. Lane L: 100 bp ladder, lane P: positive control, lane N: negative control, lanes 1 to 30 are positive samples. (**C**) Results of MAC-specific PCR from organ tissues of wild macaques. Lane L: 100 bp ladder, lane P: positive control, lane N: negative control, lanes 1 to 10 are positive samples. *M. bovis* nucleic acids was detected with primer TB1-F and TB1-R at 372 bp and *M. avium* nucleic acids was detected with primer MYCAV-R and MYCGEN-F at 180 bp.

**Figure 3 animals-11-03252-f003:**
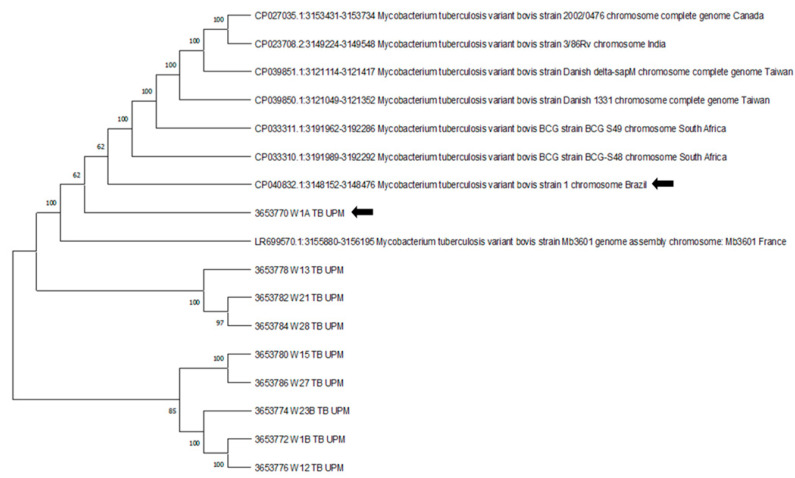
The phylogenetic tree was inferred using the UPGMA method. The optimal tree with the sum of branch length = 1.46536283 is shown. The percentage of replicate trees in which the associated taxa clustered together in the bootstrap test (1000 replicates) are shown next to the branches. The evolutionary distances were computed using the P-distance method and are in the units of the number of base differences per site. The arrow showed a close relationship between the Malaysian strain 3653770 W1A TB UPM with CP040832.1: *M. tb* variant *bovis* strain1 chromosome from Brazil.

**Table 1 animals-11-03252-t001:** Nucleotide sequence and expected product size f primers used in the PCR reaction for the detection of mycobacterial Nucleic acids in wild boar and wild macaques.

Gene	Primer Name	Nucleotide Sequence	bp	Target
16S rRNA	MYCGEN-F	AGAGTTTGATCCTGGCTCAG	1030	MTBC
	MYCGEN-R	TGCACACAGGCCACAAGGGA		
	MYCGEN-F	AGAGTTTGATCCTGGCTCAG	180	*M. avium*
	MYCAV-R	ACCAGAAGACATGCGTCTTG		
	MYCINT-F	CCTTTAGGCGCATGTCTTTA	850	*M. intracellulare*
	MYCGEN-R	TGCACACAGGCCACAAGGGA		
	TBI-F	GAACAATCCGGAGTTGACAA	372	*M. bovis*
	TB1R	AGCACGCTGTCAATCATGTA		
oxyR285	oxyRF	5′CTATGCGATCAGGCGTACTTG 3′	556	*M. bovis*
	oxyRR	5′GGT GAT ATA TCA CAC CAT A 3′		
	L1F	5′CCCGCTGATGCAAGTGCC 3′	460	*M. bovis*
	L2R	CCCGCACATCCCAACACC 3′		
Rv2073c	Rv2073cF	5′TCGCCGCTGCCAGATGAGTC 3′	600	*M. tb*
(RD9)	Rv2073cR	5′TTTGGGAGCCGCCGGTGGTGATGA3′		
hsp65631	hsp65F	5′ACC AAC GAT GGT GTG TCC AT 3′	441	MTBC
	hsp65R	5′CTT GTC GAA CCG CAT ACC CT 3′		

**Table 2 animals-11-03252-t002:** Prevalence of the diagnostic test for detection of MTBC and MAC in wild boar and wild macaques.

Diagnostic Test	Samples (N)	Positive	Percentage (95% CI)
Tuberculosis-like lesion wild boar wild macaques	30 42	9 0	30% (16.7–47.9) 0 (0–8.4)
Acid fast staining wild boar wild macaques	30 42	0 0	0 (0–11.4) 0 (0–8.4)
Mycobacterial culture Total samples	93	0	
PCR MTBC—wild boar PCR MAC—wild boar PCR MTBC— macaques	12 12 30	9 11 0	75% (46.8–91.1) 91% (64.6–98.5) 0 (0–11.4)
PCR MAC—macaques	30	10	33% (19.2–51.2)

Note: *M. bovis* nucleic acids was detected with primer TB1-F and TB1-R at 372 bp and *M. avium* nucleic acids were detected with primer MYCAV-R and MYCGEN- F at 180 bp.

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
