# Peer review of "Mycobacterium Tuberculosis and Avium Complex Investigation among Malaysian Free-Ranging Wild Boar and Wild Macaques at Wildlife-Livestock-Human Interface"

_animals, 2021, doi:10.3390/ani11113252_

Round 1

Reviewer 1 Report

This paper needs considerable work with regards to English language.  I would advise asking a colleague with more English familiarity to help edit this paper before resubmission.  In addition it will need some major revision and removal of elephant data that seem superfluous to this paper.  Further comments are attached.

Author Response

Reviewer 1:

General Comments: Overall this manuscript has merit regarding the information it is presenting

in terms of detecting evidence of mycobacterial infection or exposure in free-ranging macaques

and wild boar in the Salangor region of Malaysia. First major comment: this paper is in need of

English language revisions before resubmission. This will be important in terms of both

readability and clarification of the data presented, as many aspects of the paper are confusing

due to language issues.

Secondly, I feel that the captive elephant data, although interesting, do

not really fit with this manuscript, as the elephants are located on the other side of the country

and do not seem to be epidemiologically related to the systems in which the boar and macaques

reside. Thus the elephant data should be removed unless the authors have a good argument for

keeping them in. They seem like they belong in a separate report.

Lines 9-17: Consistent spelling of ‘’university’’

Lines 22-31: Define Acronyms or don’t use them in the Simple Summary

Lines 32-51: Abstract: Redefine all acronyms or do not use them in the abstract. There is too

much detail regarding methods in abstract (such as description of culture methods).

Lines 52-54: Perhaps add genus/species of animals in study, and Selangor to key words

Line 69: Wildlife Human Conflict area needs to be clearly defined, both regarding the acronym and how a WHC is determined. Is it an official designation of an area?

Lines 72-77: Sentences do not make sense

Line 108: acronym is wrong

Lines 150-151: clarify what type of hunting this is

Lines 152-153: Age categories should be also specified here

Lines 171-173: Why was histopathology not done?

Line 186: how much NaOH added?

Tissue Extraction Descriptions: Do not need to describe the commercial kit instructions in the

Methods if all methods are based on manufacturers Protocols

Line 214 Capitalize Mycobacterium

Lines 214-261: Were the results of multiple primers targeting the same mycobacterial species

compared?

Table 1 needs to be reformatted. It is confusing as lines do not line up etc. and labelling is

inconsistent

Line 304-310: 100% positive rates (especially in blood samples) make me suspect possible

contamination/false positives. Was this possibility accounted for and ruled out? I see the

negative controls were negative, but were these results repeatable?

Table 2 also needs reformatting

Line 330: Taxa?

Lines 372-387: Discuss why did not perform histopathology as this may have documented TBlike

lesions and acid fast staining

Line 408: agar

Lines 441-454: Isolation of organisms only refers to an organism grown in culture

Line 457: inhabit

Discussion: Perhaps talk about management approaches as well as safety precautions for

hunters in face of existence of TB in wild boar and in the context of WHC areas.

We will follow the advice to go through the English review

The reviewer comments were right, elephant data is from the National Elephant Conservation Centre (NECC), Pahang (east coast Malaysia). However, the data is included as the current study aims to demonstrate a more comprehensive surveillance of selected wild animals (both free-ranged and captive) that have potential to host mycobacterial infections. Furthermore, the initial seroprevalence study was conducted in the same centre and we try to see the current situation five years after by conducting reevaluation study.

Lines 9-17: it has been corrected accordingly

Lines 22-31: wildlife-livestock-human interface

Lines 32-51: all acronyms are mention in full

Lines 52-54: Selangor added to key words.

genus/species of animals added, wild boar (Sus scrofa), wild macaques (Macaca fascicularis) and Asian elephants (Elephas maximus).

Line 69: wildlife-livestock-human interface is the official designation

Lines 72-77:

Line 108: wildlife-livestock-human interface

Lines 150-151: registered pleasure hunters in collaboration with wildlife department officials.

Lines 152-153: Wild boar with the following age (4 yearlings, 9 subadult, and 17 adults), wild macaques with the following age (5 yearlings, 11 subadult, and 26 adults) and (7 young, 6 subadult, and 8 adults).

Lines 171-173: During the planning of our research, considering our budget constraint we decided to either take stained smear (AFS) or histopathology. We later decide to use stained smear (AFS) using Ziehl-Neelsen staining which is the method used in low resource setting to diagnose tuberculosis instead of histopathology. Furthermore, histopathological finding only as a support and not the main method for the diagnosis of TB/mycobacterial infections.

Line 186: 4.0 grams of NAOH was added in 100 mL of sterile distilled water to make a final concentration of 2% NaOH.

DNA extraction it has been reduced accordingly

Line 214 MYCAV-R and MYCGEN-F are names of forward and reverse primers used.

Lines 214-261 the reason of using different primers was to target specific members of MTBC and MAC.

Table 1 reformatted accordingly

Line 304-310 PCR from a trunk wash and blood of elephants were negative for MTBC antigens but were positive (95%) and (23%) for MAC respectively.

Table 2 reformatted accordingly

Line 330: taxa corrected accordingly

Lines 372-387: During the planning of our research, considering our budget constraint we decided to either take stained smear (AFS) or histopathology. We later decide to use stained smear (AFS) using Ziehl-Neelsen staining which is the method used in low resource setting to diagnose tuberculosis instead of histopathology. This study also decided to use the diagnosis of gross lesions (TB like lesion) compared to histopathology because studies showed gross lesions also have an acceptable sensitivity.

Line 408: agar corrected accordingly

Lines 441-454: extracted corrected accordingly

Line 457: survive corrected accordingly

Discussion added as recommended

Reviewer 2 Report

The article presents a cross-sectional study on MTC and MAC in wild boar, macaques and elephants from regions of the peninsular Malaysia. The authors performed culture with media specific to mycobacteria, microscopy, or molecular analyzes on tissue, blood or trunk wash. Most of the samples tested positive by PCR against MAC members and some wild boar samples tested positive for MTBC members. Sequencing of amplicons from positive MTBC samples was also performed and compared with known sequences.

I believe the study falls within the scope of the journal and the results are relevant enough to warrant publication. However, the manuscript needs serious improvement before it can be accepted.

Title:

The title is misleading as the manuscript does not describe a proper surveillance system (see definitions in for instance Krickeberg et al. 2012: Epidemiology, Krämer et al 2010: Modern Infectious Disease Epidemiology, or Thrusfield 2005 Veterinary Epidemiology). On the other hand, I really appreciate the honesty in the M&M section about the opportunistic character of the sampling. The results appear somewhat anecdotal, given the variability of species, regions and samples and the low sample size. Still the positive results are relevant in the context. The title also mentions semi-captive elephants. But are these captive (line 157) or semi-captive (line 5).

Introduction:

Line 69-61: This is only true in some circumstances, such as Spain, where the reference comes from.

Line 135: But the study only mentions PCR.

Materials and Methods:

Define age classes in wild boar and how these were detected.

Define when the sampling was performed. Mention at least the years, if possible also the season. This might not seem relevant for this study, but in case further samplings are done in the future, these will become important.

 Line 166 -167: Be precise, we don’t know from this sentence, whether there were 21 blood and 21 trunk wash samples, or if the sum of blood and trunk wash samples added to 21. Or simplify it as it seems that Blood samples and trunk wash were taken on every elephant of the study. Please check it.

Line 174: “tissue” is not clear. Does it refer to all samples or only organ tissue samples?

Line 179: 100x amplification is way too low to detect acid fast bacteria. For that purpose, we do not go below 1000x and use immersion oil.

Lines 182-194: The protocol only mentions organ tissues samples. How were cultures from trunk wash set up?

Line 189-194: How was the pellet used? Was it re-suspended? Also was the pellet therefore divided into one half for the L-J medium and the other half for the 7H11?

Line 198: How were tissue samples homogenized? Did the author use the same protocol as for the culture?

Line 209: The flow through what? Were the sample centrifuged in a column?

Line 215: PCR was performed on 12 tissue samples of the 30 wild boars. Which organs were tested and how were these selected? Did different sample originate from the same wild boar or were these all from separate animals? The same holds for the macaques.

Lines 219-221 and 226-227: Do these primer pairs target Mycobacterium sp. or only MTBC (compare with Table 1)?

Line 259: If I compare the panels C and D of Fig. 2 It seems that the wells were of different size.

Line 265: Which forward and reverse primers?

Results

Line 288: Even if the author can calculate a percentage of positivity, the sampling method and low sample size hardly warrant the estimation of a somewhat reliable prevalence. This for the same reason as the author cannot pretend to have performed a surveillance.

Line 293-295: Did some animals show multiple lesions? How many of them?

Figure 1 both panels are out of focus, try to find better pictures if you have.

Line 306: The samples were positives to MTBC or M. bovis (compare with Table 2).

Line 308: The author tested for nucleic acids not antigens.

Line 308: There were not 100% positive but 95% (compare with Table 2).

Table 2; Remove “apparent prevalence”.

Table2: It seems that results are missing (MTBC elephants).

Line 328: The figure shows a phylogenetic tree, not the evolutionary history of Mycobacteria.

Discussion

Line 339-344: The comparison is irrelevant as the management probably completely different.

Line 350 and 352: scavenging is a form of digestive route.

Paragraph at line 358: Results appear in this paragraph, they should be moved to the result section.

Line 359-361: was the cavitation in a lung or in a submandibular LN?

Line 361-362: Show the age distribution of the sampled animals? Is there a sampling bias toward older age categories? Also depending on the definition of the age classes, the adult class probably comprises a larger range of ages.

Paragraph at line 401: Dos it mean that the negative culture could have been due to killing of the bacteria during the treatment?

Line 415: So PCR was performed on organs with and without lesions?

Line 443-446: The reference comes from Finland and refers to M. avium subsp. hominisuis only. The author cannot conclude from it that Malaysian wild boar are important for the transmission of all MAC members.

Line 452-454: The authors could develop this idea. Given the high percentage of animals positive to MAC, molecular studies on these could help identifying potential inter-species transmission for instance.

I am not a native English speaker myself but I still believe that the manuscript needs serious language improvement. Here are some examples but I haven’t checked the whole text.

Line 3: Replace “free-range” with “free-ranging”.

Line 69: Define “WHC” at first appearance.

Line 77: Replace “this species” with “these species”.

Line 88: Replace “suggestive” with “suggesting that”.

Line 95: Replace “eventually” with “potentially”.

Line 107: The abbreviation “ASF” is not repeated, delete it.

Line 111-113: I don’t understand this sentence, rewrite it.

Line 115-115: The sentence is verbose, simplify it. For instance: “The NAATs techniques have improved TB diagnosis compared to mycobacterial culture”.

Line 118-119: The sentence can be removed without losing important information.

Line 124: Remove “but”

Line 125: Replace “tissue” with “sample”.

Line 165: Remove “the”.

Line 166: Replace “in” with “at”.

Line 192: Use uppercase “O” in CO2.

Line 202: Replace “is” with “was”.

Line 229: Replace “is” with “was”.

Line 234: Format the reference correctly

Line 233-242: Does this protocol refer to the M. avium PCR? I guess so, given the positive control and the fact that the next paragraph describes the PCR of the Elephants and Macaques. Please check it. Also what were the PCR conditions for the wild boar?

Line 280: Replace “is” with “was”.

Line 288: Replace “was” with “were”.

Line 295: Replace “all” with “any”.

Line 297-298: Reorganize the phrase, for instance “Culture of tissues wild boar and macaques, and

trunk wash from elephants[…]”.

Line 353: Replace “infection” with “infected”.

Line 395-396: Rephrase, if something is vital, it is certainly for the survival.

Author Response

Reviewer 2:

Title:

The title is misleading as the manuscript does not describe a proper surveillance system (see definitions in for instance Krickeberg et al. 2012: Epidemiology, Krämer et al 2010: Modern Infectious Disease Epidemiology, or Thrusfield 2005 Veterinary Epidemiology). On the other hand, I really appreciate the honesty in the M&M section about the opportunistic character of the sampling. The results appear somewhat anecdotal, given the variability of species, regions and samples and the low sample size. Still the positive results are relevant in the context. The title also mentions semi-captive elephants. But are these captive (line 157) or semi-captive (line 5).

Introduction:

Line 69-61: This is only true in some circumstances, such as Spain, where the reference comes from.

Line 135: But the study only mentions PCR.

Materials and Methods:

Define age classes in wild boar and how these were detected.

Define when the sampling was performed. Mention at least the years, if possible also the season. This might not seem relevant for this study, but in case further samplings are done in the future, these will become important.

 Line 166 -167: Be precise, we don’t know from this sentence, whether there were 21 blood and 21 trunk wash samples, or if the sum of blood and trunk wash samples added to 21. Or simplify it as it seems that Blood samples and trunk wash were taken on every elephant of the study. Please check it.

Line 174: “tissue” is not clear. Does it refer to all samples or only organ tissue samples?

Line 179: 100x amplification is way too low to detect acid fast bacteria. For that purpose, we do not go below 1000x and use immersion oil.

Lines 182-194: The protocol only mentions organ tissues samples. How were cultures from trunk wash set up?

Line 189-194: How was the pellet used? Was it re-suspended? Also was the pellet therefore divided into one half for the L-J medium and the other half for the 7H11?

Line 198: How were tissue samples homogenized? Did the author use the same protocol as for the culture?

Line 209: The flow through what? Were the sample centrifuged in a column?

Line 215: PCR was performed on 12 tissue samples of the 30 wild boars. Which organs were tested and how were these selected? Did different sample originate from the same wild boar or were these all from separate animals? The same holds for the macaques.

Lines 219-221 and 226-227: Do these primer pairs target Mycobacterium sp. or only MTBC (compare with Table 1)?

Line 259: If I compare the panels C and D of Fig. 2 It seems that the wells were of different size.

Line 265: Which forward and reverse primers?

Results

Line 288: Even if the author can calculate a percentage of positivity, the sampling method and low sample size hardly warrant the estimation of a somewhat reliable prevalence. This for the same reason as the author cannot pretend to have performed a surveillance.

Line 293-295: Did some animals show multiple lesions? How many of them?

Figure 1 both panels are out of focus, try to find better pictures if you have.

Line 306: The samples were positives to MTBC or M. bovis (compare with Table 2).

Line 308: The author tested for nucleic acids not antigens.

Line 308: There were not 100% positive but 95% (compare with Table 2).

Table 2; Remove “apparent prevalence”.

Table2: It seems that results are missing (MTBC elephants).

Line 328: The figure shows a phylogenetic tree, not the evolutionary history of Mycobacteria.

Discussion

Line 339-344: The comparison is irrelevant as the management probably completely different.

Line 350 and 352: scavenging is a form of digestive route.

Paragraph at line 358: Results appear in this paragraph; they should be moved to the result section.

Line 359-361: was the cavitation in a lung or in a submandibular LN?

Line 361-362: Show the age distribution of the sampled animals? Is there a sampling bias toward older age categories? Also depending on the definition of the age classes, the adult class probably comprises a larger range of ages.

Paragraph at line 401: Dos it means that the negative culture could have been due to killing of the bacteria during the treatment?

Line 415: So PCR was performed on organs with and without lesions?

Line 443-446: The reference comes from Finland and refers to M. avium subsp. hominisuis only. The author cannot conclude from it that Malaysian wild boar is important for the transmission of all MAC members.

Line 452-454: The authors could develop this idea. Given the high percentage of animals positive to MAC, molecular studies on these could help identifying potential inter-species transmission for instance.

 I am not a native English speaker myself but I still believe that the manuscript needs serious language improvement. Here are some examples but I haven’t checked the whole text.

Line 3: Replace “free-range” with “free-ranging”.

Line 69: Define “WHC” at first appearance.

Line 77: Replace “this species” with “these species”.

Line 88: Replace “suggestive” with “suggesting that”.

Line 95: Replace “eventually” with “potentially”.

Line 107: The abbreviation “ASF” is not repeated, delete it.

Line 111-113: I don’t understand this sentence, rewrite it.

Line 115-115: The sentence is verbose, simplify it. For instance: “The NAATs techniques have improved TB diagnosis compared to mycobacterial culture”.

Line 118-119: The sentence can be removed without losing important information.

Line 124: Remove “but”

Line 125: Replace “tissue” with “sample”.

Line 165: Remove “the”.

Line 166: Replace “in” with “at”.

Line 192: Use uppercase “O” in CO2.

Line 202: Replace “is” with “was”.

Line 229: Replace “is” with “was”.

Line 234: Format the reference correctly

Line 233-242: Does this protocol refer to the M. avium PCR? I guess so, given the positive control and the fact that the next paragraph describes the PCR of the Elephants and Macaques. Please check it. Also what were the PCR conditions for the wild boar?

Line 280: Replace “is” with “was”.

Line 288: Replace “was” with “were”.

Line 295: Replace “all” with “any”.

Line 297-298: Reorganize the phrase, for instance “Culture of tissues wild boar and macaques, and

trunk wash from elephants […]”.

Line 353: Replace “infection” with “infected”.

Line 395-396: Rephrase, if something is vital, it is certainly for the survival.

Spatial Surveillance (Krämer et al 2010): Modern Infectious Disease Epidemiology,

Surveillance targets wildlife populations classified as healthy to demonstrate the absence of infection (OIE, 2011).

(line 157) Semi-captive

Line 69-61: There were reports from other countries e.g Italy, France, Switzerland and south Korea. lovane et al., 2020; Richomme et al., 2013; Beerli et al., 2015; Jang et el.,2017.

Line 135: Tuberculosis like lesion, Acid fast stained, Mycobacterial culture and PCR. Table 2

The age of the wild boar was established using a tooth eruption pattern (Sáez-Royuela et al., 1989), and the animals were divided into three age categories: yearling (0–12 months); sub-adults (13–36 months); and adults (>36 months).

Sampling was performed from April to August 2019-2020.

Line 166 -167: Blood samples and trunk wash were taken on every elephant of the study. making 21 blood and 21 trunk wash samples.

Line 174: From each of the organs, tissues were collected and stored in -20°C for further analysis.

Line 179: at 1000x using immersion oil

Lines 182-194: For the trunk wash protocol was as described by (Ong et al., 2013).

Line 189-194: the pellet was inoculated on Lowenstein Jensen (L-J) media and Middlebrook 7H11 using sterile swab stick.

Line 198: tissue samples were homogenised using stomacher in the lab.

Line 209: centrifuged in 50ml falcon tubes.

Line 215: organs with Tuberculosis like lesion TBLL were selected for PCR. samples originate from different wild boars and macaques.

Lines 219-221 and 226-227 these primers were mainly to target MTBC members and MAC.

Line 259: Yes panels of gel caster C is 30 samples and panels of D is 16 samples.

Line 265: amended accordingly

Line 288

Line 293-295: yes 11 wild boars have multiple lesions

Figure 1 no better pictures than this one.

Line 306: these samples were positive for M. bovis because primers used were specific to M.bovis

Line 308: nucleic acids

Line 308: were positive (95%) and (23%) for MAC.

Table 2; removed as recommened

Table 2; amended accordingly all result are inserted in table.

Line 328: phylogenetic tree

Line 339-344: yes, agreed different wildlife setting but we are trying to prove the existence of TB in these species of animals as reported in other part of the world

Line 350 and 352: Localized lesions could be due to infection transmitted through oral route by scavenging of tuberculous carrion such as TB dead animal.

Paragraph at line 358: corrected accordingly

Line 359-361: cavitation obtained from submandibular LN

Line 361-362: yes, there is sampling bias but not deliberate because its opportunistic sampling

Paragraph at line 401: it’s possible because of the few lesions encountered in the organs during post-mortem

Line 415: PCR was performed on those with lesions and without lesions.

Line 443-446: just trying to indicates that local wild boar are important for possible MAC transmission.

Line 452-454: Noted

Noted

Line 3: corrected accordingly free-ranging”.

Line 69: wildlife-livestock-human interface

Line 77: corrected accordingly these species

Line 88: corrected accordingly suggesting that

Line 95: corrected accordingly potentially

Line 107: deleted as recommended.

Line 111-113: corrected accordingly

Line 115-115: corrected accordingly

Line 118-119: deleted as recommended

Line 124: deleted as recommended

Line 125: Replaced as recommended

Line 165: deleted as recommended

Line 166: Replaced as recommended

Line 192: Replaced as recommended

Line 202 Replaced as recommended

Line 229: Replaced as recommended

Line 234: corrected accordingly

Line 233-242: the protocol varies depending on the primers used. some primers protocol is for M.bovis  in wild boars while others are for M.tb in macaques and elephants.

Line 280: Replaced as recommended

Line 288: Replaced as recommended

Line 295: Replaced as recommended

Line 297-298: amended as recommended

Line 353: Replaced as recommended

Line 395-396: amended as recommended

Round 2

Reviewer 1 Report

This paper is greatly improved regarding language issues.  There are still issues within the manuscript, but I have specified them in my comments.

I continue to feel that the elephant data are not relevant to this manuscript. These data should be removed and written up in a separate report.  If you intend to follow up investigations on the elephants in five years, you should then reference the earlier report, or report all the data in one paper at that time.

Author Response

Reviewer 1:

 Title is now misleading because it implies that spatial data are involved in this study, and there are not.

I will maintain that the elephant data are not appropriate for this study. They are in a different region and are captive (or semi-captive?).

Line 27: antigen

Line 28: 75% of samples or animals from which samples were taken?

Line 33: Remove “animals” or write “wild animals”

Line 33: remove “as”

Line 37: wash

Lines 48-52: Lymphoid tissues from twelve animals? This is not clearly written here.

Line 49: Either lymphoid tissues or submandibular lymphoid tissue samples. As written it is redundant

Line 53: You switched to using wildlife-livestock-human interface so this should be reflected in the key words as well. Conflict zone is no longer a key word in this paper

Line 61: Either “wildlife” or “wild animals”

Line 62: Wild boar can act as maintenance hosts for TB. This has not necessarily been demonstrated in your study.

Line 97: As above “wildlife” or “wild animals”

Line 104: “conflict areas”

Line 111: Acronym is wrong

Line 125: As above “wildlife” or “wild animals”

Line 130: “mycobacteria”

Lines 134-138: these data belong in another report as a follow up to the last one mentioned here.

Lines 152-153: Remove “and their”

Lines 152-156: This sentence is run on and needs to be separated into two sentences

Lines 160-165: Remove

Lines 177-181: Is this intended as a type of ranking? In the article referenced, these terms were simply definitions of lesion descriptions. Better to say that the lesions were categorized base on the descriptions in Crashaw et al.

Line 193: 2% NaOH

Line 199: “…sticks, and…”

Line 200: start new sentence here

Lines 202-208: Remove

Lines 215-217: Again, how many animals do these tissues represent?

Line 219: “primers”

Line 222-225: This is not a complete sentence. And do all these primer sets detect M. bovis?

Lines 237-253: Were there no other controls besides M. avium avium? What about the other species you were targeting?

Line 299: “They were observed…”

Lines 310-316: Since you used multiple sets of primers to identify M. bovis, did you test these tissues using all of the primers, and did the results agree?

Lines 442-452: How did a Brazilian strain get introduced into Malaysian wildlife? Is there a livestock population in the area that is also host to this particular strain?

Line 499: Remove “animals”

Spatial; it was removed from title as recommended

All captive Asian Elephant data were removed as recommended

Line 27: antigen; Corrected accordingly

Line 28: lymph node samples

Line 33: removed as recommended

Line 33: removed as recommended

Line 37: wash; corrected

Lines 48-52: yes, Lymphoid tissues from twelve animals.

Line 49: Corrected as submandibular lymphoid tissues samples

Line 53: Keywords; WLHI wildlife-livestock-human interface Corrected accordingly

Line 61: wild animals; Corrected accordingly

Line 62: yes

Line 97: wild animals; Corrected accordingly

Line 104: wildlife-livestock-human interface Corrected accordingly

Line 111: PCR; Corrected accordingly

Line 125: wild animals; Corrected accordingly

Line 130: “mycobacteria; Corrected accordingly

Lines 134-138: Similar research was previously reported in the same study area but they are different project.

Lines 152-153: removed as recommended

Lines 152-156: the sentences are separated accordingly

Lines 160-165: elephant data removed as recommended

Lines 177-181: Corrected accordingly

Line 193: yes 2% NaOH

Line 199: Corrected accordingly

Line 200: amended accordingly

Lines 202-208: removed as recommended

Lines 215-217: Conventional PCR on 12 lymphoid tissue samples of wild boar, and 10 tissue samples of macaques was conducted

Line 219: primers, Corrected accordingly

Line 222-225: the Primers were set to detect major members of MTBC which are M.bovis and M.tb while others detect M.avium.

Lines 237-253: for targeting MTBC members, M. tuberculosis H37R3 (ATCC® 25177TM) was used as positive control while for MAC members, Mycobacterium avium subspecies avium Chester (ATCC® 15769TM), was used as positive control.

Line 299:

Lines 310-316: Among the primers tested only TB1-F and TB1-R was able to detect M. bovis. while for MAC the primer MYCAV-R and MYCGEN-F was able to detect M. avium in all tissues tested.

Lines 442-452: there are livestock farms in sampling areas and some are imported from other regions and farmers are operating in their traditional way of open house and extensive livestock farming this will bring interaction, thereby facilitating the transmission of tuberculosis between wildlife and livestock.

Line 499: removed as recommended

Reviewer 2 Report

I acknowledge the fact that the authors made efforts to answer my comments and those of the reviewer 1 to the first version of the manuscript. However, I believe it is still not ready to be published yet

Remove the word “surveillance” throughout the manuscript and especially in the title. As I wrote in my first review, this study does not fulfill the requirements to be called a surveillance. More precisely, the sampling is neither ongoing nor systematic. I am seriously concerned by the title change from “surveillance” to “spatial surveillance” which is even more misleading. The analysis has no spatial component at all. In other words, the title does not reflect the work that was done.

I support the view of reviewer 1 about removing the elephant part from the manuscript. These animals originate from another region, they are captives, and they were already tested positive in an earlier study. The latter is in clear contradiction with the first line of the discussion, which underlines the novel character of this study. The elephant part is a follow-up study that could well be published as a short communication.

The comment of reviewer 1 about the wildlife human conflict area (WHC) has not been properly addressed. 1) The authors simply changed the expression “wildlife human conflict area” with “wildlife-livestock-human interface” without giving any definition. 2) I agree that the latter expression is commonly used when referring to any situation where direct or indirect disease transmission may occur between human, domestic animal, or wildlife. But it is not always correctly applied in this manuscript. For instance, the paragraph at lines 93-108 attempted to define a conflict area. Now it makes little sense to use the expression wildlife-livestock-human interface. 3) The second expression implies the involvement of livestock in the study area, which is found nowhere in the manuscript. 4) Consequently, both expressions cannot be used as synonyms, it is as simple as they read: the first refers to areas where interests of humans and wildlife are not compatible and the second refers to situations with direct or indirect contact between human, livestock, and wildlife. But nowhere does the manuscript describe the kind of conflict that can occur between human and wildlife in the study area, nor does it describe any situation with potential disease transmission of infectious agents specific to the study area. The authors should decide which expression fits best to their study area, define it, and illustrate with clear examples specific to their area and not only with examples from the literature which might not necessarily apply there.

The opportunistic sampling and small sample size only allow vague inference about the population at risk. Remove most of the word “prevalence” replace them with “percentage”. Also use the 95%CI only sparingly as it is just as misleading as the word “prevalence”.

Line 27: Remove “antigen detection”. The method does not mention any antigen detection method. Remove it throughout the text.

Line 62: Add “may” after wild boar. The authors answered to my comment citing Beerli et al., 2015 among others. But the former study just showed how unlikely the wild boar seems to play a role in the epidemiology of bovine TB in Switzerland.

Figure 321: What do the arrow show? Explain it and make them more visible, I missed them during the first review. Or if they are not important, remove them.

Line 330: What is the “amplified fragments of DNA product sequence result"? Do the authors mean the sequencing of the amplicon?

Line 352: I agree that the uncommon hunting of wild boar in Malaysia may explain the small sample size in this study compared to the former ones. But the low percentage of lesions, if these are due to MTBC, is better explained through different wild boar management practices when compared to the hot spots in Mediterranean Spain and Portugal that the authors refer to.

Line 480: “management approaches […] will help”. This is what we hope to see in Spain but are these measures planned in Malaysia? Probably not. So if it is not planned this discussion is not relevant for this study.

The authors should pay attention to the classification of mycobacteria and pay attention to when they should use the word species, complex, or genus. More precisely:

Line 219-221: Check the reference, the primer pair MYCGEN-F and MYCGEN-R allows the detection of only members of the MTBC, not all Mycobacteria sp. Rephrase it.

Line 226-227: Check the reference, the primer pair hsp65F and hsp65R allows the detection of only members of the MTBC, not all Mycobacteria sp. Rephrase it.

Table 1: Remove “Genus” at both “Genus MTBC”. A complex is simply not a genus.

Table 1: “hsp65631”, 631 should be superscript.

Line 311: “…PCR was only done for 12 wild boars with TBLLs…”, correct the numbers or the sentence, line 294 and table 2 show that there were 9 wild boars with TBLLs, not 12.

MAC is also a complex and not a single bacterium (see line 453 and 459: “[MAC] was detected”, line 454: “MAC infection can occur”, line 460 “MAC has been extracted”, Line 467: “MAC is known to cause”). All these inaccuracies make me think that the authors did not understand the classification of Mycobacterium spp. MAC comprise M. avium and all subspecies such as subsp. paratuberculosis or subsp. hominisuis, but also many other species. The authors should rephrase the text to make this clear. Consequently, some statements like the line 456-458 are misleading. Like this one would understand that the authors assume that the wild boar were infected with subsp. hominisuis, which is no known yet. As long as no further typing is done, the exact member (or maybe members) of MAC present in wild boar remains unknown.

There was an attempt to improve the English language. But clear improvements are still necessary, and I strongly recommend the authors to ask a native speaker to check the text. Here are some examples of possible language improvement:

Line 4: “in” should be “at”.

Line 36: “range” should be “ranging”, check it throughout the text as I mentioned it in my first review.

Line 37: “washed” should be “wash”.

Line 38: “stain” should be “staining”.

Line 47: “a single nodular lesion was” should be “single nodular lesions were”.

Line 79: “and” should be “or”, unless Tb in NHP is always caused by the three species of mycobacteria.

Line 109: remove either “diagnosis” or “detection”.

Line 113: “for growth” should be “to grow”.

Line 148: “from” should be “by”, unless the samples were taken on the hunters and the wildlife officials.

Line 168: “by a venipuncture from” should be “by venipuncture of”.

Line 172: This sentence is redundant, remove it.

Line 294: “was” should be “were”.

Line 313-216: This is not a phrase, rewrite it.

Table 2: “stained” should be “staining”.

Line 345: “in” should be “at”.

Line 347: replace “higher gross TBLL was” with “higher values were” or something similar.

Any verb describing something in the past should be in the past form for instance:

Lines 135: “is” should be “was”

line 150: “are” should be “were”.

Minor changes:

Line 225: “M. tb” in italic is a weird abbreviation. Simplify it to Mtb or write out throughout the text.

Table 2: remove the parenthesis to the species names.

Line 368: Change “TB lesions” with “TBLLs”.

Line 385: “test” should be “staining”.

Author Response

Reviewer 2:

Remove the word “surveillance” throughout the manuscript and especially in the title. As I wrote in my first review, this study does not fulfill the requirements to be called a surveillance. More precisely, the sampling is neither ongoing nor systematic. I am seriously concerned by the title change from “surveillance” to “spatial surveillance” which is even more misleading. The analysis has no spatial component at all. In other words, the title does not reflect the work that was done.

I support the view of reviewer 1 about removing the elephant part from the manuscript. These animals originate from another region, they are captives, and they were already tested positive in an earlier study. The latter is in clear contradiction with the first line of the discussion, which underlines the novel character of this study. The elephant part is a follow-up study that could well be published as a short communication.

The comment of reviewer 1 about the wildlife human conflict area (WHC) has not been properly addressed. 1) The authors simply changed the expression “wildlife human conflict area” with “wildlife-livestock-human interface” without giving any definition. 2) I agree that the latter expression is commonly used when referring to any situation where direct or indirect disease transmission may occur between human, domestic animal, or wildlife. But it is not always correctly applied in this manuscript. For instance, the paragraph at lines 93-108 attempted to define a conflict area. Now it makes little sense to use the expression wildlife-livestock-human interface. 3) The second expression implies the involvement of livestock in the study area, which is found nowhere in the manuscript. 4) Consequently, both expressions cannot be used as synonyms, it is as simple as they read: the first refers to areas where interests of humans and wildlife are not compatible and the second refers to situations with direct or indirect contact between human, livestock, and wildlife. But nowhere does the manuscript describe the kind of conflict that can occur between human and wildlife in the study area, nor does it describe any situation with potential disease transmission of infectious agents specific to the study area. The authors should decide which expression fits best to their study area, define it, and illustrate with clear examples specific to their area and not only with examples from the literature which might not necessarily apply there.

The opportunistic sampling and small sample size only allow vague inference about the population at risk. Remove most of the word “prevalence” replace them with “percentage”. Also use the 95%CI only sparingly as it is just as misleading as the word “prevalence”.

Line 27: Remove “antigen detection”. The method does not mention any antigen detection method. Remove it throughout the text.

Line 62: Add “may” after wild boar. The authors answered to my comment citing Beerli et al., 2015 among others. But the former study just showed how unlikely the wild boar seems to play a role in the epidemiology of bovine TB in Switzerland.

Figure 321: What do the arrow show? Explain it and make them more visible, I missed them during the first review. Or if they are not important, remove them.

Line 330: What is the “amplified fragments of DNA product sequence result"? Do the authors mean the sequencing of the amplicon?

Line 352: I agree that the uncommon hunting of wild boar in Malaysia may explain the small sample size in this study compared to the former ones. But the low percentage of lesions, if these are due to MTBC, is better explained through different wild boar management practices when compared to the hot spots in Mediterranean Spain and Portugal that the authors refer to.

Line 480: “management approaches […] will help”. This is what we hope to see in Spain but are these measures planned in Malaysia? Probably not. So if it is not planned this discussion is not relevant for this study.

The authors should pay attention to the classification of mycobacteria and pay attention to when they should use the word species, complex, or genus. More precisely:

Line 219-221: Check the reference, the primer pair MYCGEN-F and MYCGEN-R allows the detection of only members of the MTBC, not all Mycobacteria sp. Rephrase it.

Line 226-227: Check the reference, the primer pair hsp65F and hsp65R allows the detection of only members of the MTBC, not all Mycobacteria sp. Rephrase it.

Table 1: Remove “Genus” at both “Genus MTBC”. A complex is simply not a genus.

Table 1: “hsp65631”, 631 should be superscript.

Line 311: “…PCR was only done for 12 wild boars with TBLLs…”, correct the numbers or the sentence,

 line 294 and table 2 show that there were 9 wild boars with TBLLs, not 12.

MAC is also a complex and not a single bacterium (see line 453 and 459: “[MAC] was detected”, line 454: “MAC infection can occur”, line 460 “MAC has been extracted”, Line 467: “MAC is known to cause”). All these inaccuracies make me think that the authors did not understand the classification of Mycobacterium spp. MAC comprise M. avium and all subspecies such as subsp. paratuberculosis or subsp. hominisuis, but also many other species. The authors should rephrase the text to make this clear. Consequently, some statements like the line 456-458 are misleading. Like this one would understand that the authors assume that the wild boar were infected with subsp. hominisuis, which is no known yet. As long as no further typing is done, the exact member (or maybe members) of MAC present in wild boar remains unknown.

There was an attempt to improve the English language. But clear improvements are still necessary, and I strongly recommend the authors to ask a native speaker to check the text. Here are some examples of possible language improvement:

Line 4: “in” should be “at”.

Line 36: “range” should be “ranging”, check it throughout the text as I mentioned it in my first review.

Line 37: “washed” should be “wash”.

Line 38: “stain” should be “staining”.

Line 47: “a single nodular lesion was” should be “single nodular lesions were”.

Line 79: “and” should be “or”, unless Tb in NHP is always caused by the three species of mycobacteria.

Line 109: remove either “diagnosis” or “detection”.

Line 113: “for growth” should be “to grow”.

Line 148: “from” should be “by”, unless the samples were taken on the hunters and the wildlife officials.

Line 168: “by a venipuncture from” should be “by venipuncture of”.

Line 172: This sentence is redundant, remove it.

Line 294: “was” should be “were”.

Line 313-216: This is not a phrase, rewrite it.

Table 2: “stained” should be “staining”.

Line 345: “in” should be “at”.

Line 347: replace “higher gross TBLL was” with “higher values were” or something similar.

Any verb describing something in the past should be in the past form for instance:

Lines 135: “is” should be “was”

line 150: “are” should be “were”.

Minor changes:

Line 225: “M. tb” in italic is a weird abbreviation. Simplify it to Mtb or write out throughout the text.

Table 2: remove the parenthesis to the species names.

Line 368: Change “TB lesions” with “TBLLs”.

Line 385: “test” should be “staining”.

Titled has been modified to

Mycobacterium tuberculosis and avium Complex Investigation Among Malaysian Free-Ranging Wild Boar and Wild Macaques at Wildlife–Livestock–Human Interface

All data relating to Asian elephant is removed.

1) The (WLHI) represent a critical point for cross species transmission and emergence of pathogens

The state of Selangor (sampling area) is a typical example which has a number of high-risk areas for zoonotic TB in domestic cattle and human, due to the number of dairy cattle farms and the diversity of wildlife and encroachment of wildlife into human settlement.

the word “prevalence is removed as recommended

Line 27: nucleic acids

Line 62: added as recommended

Figure 321: The arrow showed a close relationship between the Malaysian strain 3653770 W1A TB UPM with CP040832.1: M. tb variant bovis strain1 chromosome from Brazil.

Line 330:it means the sequencing of the amplicon

Line 352: low percentage of lesions, maybe as result of taking samples from apparently healthy or not chronically diseased wild boars and tuberculosis is not endemic in Malaysian wildlife.

Line 480: these measures are not planned in Malaysia

noted.

Line 219-221: yes, MYCGEN-F and MYCGEN-R primer pair allows the detection of only members of the MTBC. corrected accordingly 

Line 226-227: yes hsp65F and hsp65R primer pair allows the detection of only members of the MTBC. corrected accordingly 

removed accordingly 

superscript. added as recommended

Line 311: PCR was only done for 12 wild boars (9 TBLL and 3 non TBLL)

line 294: Yes Table 2 only shows 9 positive visible lesions or TBLL

453 and 459: amended accordingly

line 454: amended accordingly

line 460: amended accordingly

Line 467: amended accordingly

Line 4: at; corrected accordingly

Line 36: ranging; added accordingly

Line 37: removed

Line 38: staining; corrected accordingly

Line 47: single nodular lesions were; corrected accordingly

Line 79: should be; corrected accordingly

Line 109: diagnosis; corrected accordingly

Line 113: grow; corrected accordingly

Line 148: by; corrected accordingly

Line 168: by venipuncture of”; corrected accordingly

Line 172: The sentence has been improved

Line 294: were; corrected accordingly

Line 313-216:

Table 2: staining; corrected accordingly

Line 345:at; corrected accordingly

Line 347: higher values were; corrected accordingly

Lines 135: corrected accordingly

line 150: corrected accordingly

Line 225: corrected accordingly

Table 2: parenthesis to the species names removed as recommended

Line 368: TBLLs corrected as recommended

Line 385:staining corrected accordingly